# A Game-Theoretic Approach for Network Security Using Honeypots

**Răzvan Florea** [1,*] and **Mitică Craus** [2]

1   Faculty of Automatic Control and Computer Engineering, "Gheorghe Asachi" Technical University of Iasi, 700050 Iasi, Romania
2   Department of Computer Science and Engineering, Faculty of Automatic Control and Computer Engineering, "Gheorghe Asachi" Technical University of Iasi, 700050 Iasi, Romania
*   Correspondence: razvan.florea@student.tuiasi.ro

**Abstract:** Cybersecurity plays an increasing role in today's digital space, and its methods must keep pace with the changes. Both public and private sector researchers have put efforts into strengthening the security of networks by proposing new approaches. This paper presents a method to solve a game theory model by defining the contents of the game payoff matrix and incorporating honeypots in the defense strategy. Using a probabilistic approach we propose the course-of-action Stackelberg game (CoASG), where every path of the graph leads to an undesirable state based on security issues found in every host. The reality of the system is represented by a cost function which helps us to define a payoff matrix and find the best possible combination of the strategies once the game is run. The results show the benefits of using this model in the early prevention stages for detecting cyberattack patterns.

**Keywords:** cybersecurity; game theory; honeypot; cyberdeception; attack graph

## 1. Introduction

Cybersecurity issues are vastly discussed and analyzed by various specialists and researchers from all over the world. Moreover, the pandemic situation decided a lot of companies to use cloud infrastructures more often than before and accommodate their employees to use the new facilities to work from home. In addition, governments and public institutions were forced to implement efficient online solutions to provide services for the population.

This study focuses on the defender who improves network security by introducing honeypots. Even if the cost of honeypots depends on the performance and applicability, their existence does not interfere with legitimate users but acts as a decoy and distracts attackers from the real virtual machines and can send alarms to the administrator in case of a breach. From the mindset of an attacker, we can also understand that an experienced person should anticipate the existence of honeypots.

Cyberdeception is used nowadays by both parties, the defender and the attacker. On their side, the defender can use some techniques to give false beliefs to the attacker by using decoys, false information or trying to camouflage the network. These techniques are very similar to those used in military conflicts and can be applied in the cyberworld. On their side, the attacker is using deception and different strategies to protect their identity or let the defender think they are a legitimate user or a service. Moreover, the attacker can use deception techniques in order to gather information about potential victims. Decoys and deceptive signals are mandatory for manipulating the opponent and creating a false perception of reality so the attacker using reconnaissance tools gathers only misleading information and wastes their time.

In his work [1], Almeshekah stated that human beings are not very keen on detecting deception and the studies run on different classes of workers or students led to the fact that

over 45 % of the subjects did not detect deception. Furthermore, Whaley, in his book [2], stated that the deceiver was almost always successful, regardless of the sophistication of their victim in the same art.

The best definition which is widely accepted by authors such as Almeshekah [1] is the one proposed by Yuill [3] where cyberdeception refers to all planned actions taken to mislead attackers and to cause them to take or not to take actions that help cybersecurity defense.

John Boyd presented in his work [4] the OODA loop, which means to observe, orient, decide and act, as a cyclic process model in order for the defender to be able to assess an event and add an input for the creation of a common operational picture. The cybersecurity game between a defender and an attacker can be seen as an OODA loop race and the winner is decided by who executes this loop faster. Using honeypots, a defender has an advantage in front of the attacker because all the decoys are affecting the attacker's loop.

## 2. Related Work

In [5], Yoon used moving-target defense in software defined networks in order to tackle a potential attacker who is trying to gain access from an outside network. They used a directed acyclic graph but introduced a shuffling technique by allowing the SDN to change the IP of hosts, by network shuffling or by a packet header randomization. The aim of this technique was to enhance the SDN by controlling the shuffling process and was not using honeypots.

Unfortunately, there is no standard format for attack graphs and Lallie [6] conducted a survey on over 180 attack graphs trying to identify the visual syntax of those graphs and concluded that even if they were very popular used, there must be conducted more studies to standardize cybergraphs. The same objective of providing a survey for standardization was realized by Zeng [7], who concluded the same thing as Lallie [6]—the necessity of standardization.

On the other side, the attacker can use some techniques to identify honeypots and Huang [8] introduced an artificial intelligence method to identify honeypots; however, our model is focused on the defender.

Khouzani [9] described a cybersecurity optimization problem where the model was based on a minimax optimization problem and the same approach was considered by us, where the defender problem is to keep in mind the reaction of the attacker, but the model proposed is very close to the category known as "network interdiction problems".

Lippmann [10] realized a survey on the generation and analysis techniques of attack graphs and made a strong point of reviewing the attack graph technique. Some authors [11] realized a survey on different attack graph techniques such as the *state-enumeration-based approach*, *TVA approach*, *logic-programming-based approach* or the *netSPA Approach* and emphasized the still-unexplored area and the potential of using attack graph in network security.

## 3. Materials and Methods
### 3.1. Attack Graphs: A Model for Risk Analysis

In our model, we assume the defender uses several honeypots *h* and tries to deceive the attacker by doubling the most important hosts such as SQL databases, web servers, etc. More honeypots increase the chances that the attacker will access a honeypot and not a real host and the cost they pay will be higher. Furthermore, the alerts transmitted during the attack can provide time for the defender to take real actions to stop the attack and therefore pay a lower cost.

The exploits are taken into consideration so that an attacker is searching the ones for gaining privileged access to the target. We assume that rewards with low costs can be obtained by an attacker if a target can be successfully exploited. Security vulnerabilities are entrances for malicious exploiters in the operating system, application, hardware or other software. The failure in implementing security policies or lacking security system updates/patches can allow a malicious person to gain access to a system.

The attack graph model with the strongest ability in the description of the network attack process is the one proposed by Swiler et al. [12]. The nodes of the attack graph represent stages of an attack, and the edges represent an attack that changes the state. In general, the nodes of the attack graph look like nodes of the attack instantiated with particular users and machines. The edges of the graph are labeled by a probability-of-success (or cost) measure and a documentation string for the user interface.

### 3.2. Attack Graph Framework

The attack graph framework is simply designed to collect information about the network such as topology, vulnerabilities, configuration and connectivity so that the information collected is used to generate the attack graph.

To acquire network information, there is a need to have network access from a source to a destination. In this way, a reachability matrix can be modeled which is a two-dimensional matrix where the source IP is the first dimension, and the destination IP is the second dimension.

The reachability network's matrix described in Table 1 defines the VM as a virtual machine that is hosted on a server, the IP that is allocated to the VM, and the services that are running on a VM which allow interaction with other services running on different VMs.

**Table 1.** Reachability matrix.

|  |  |  | Destinations | | |
|---|---|---|---|---|---|
|  |  |  | $VM_x$ $IP_x$ | $VM_y$ $IP_y$ | $VM_z$ $IP_z$ |
| Sources | $VM_a$ | $IP_a$ | $Service_m$ |  | $Service_o$ |
|  | $VM_b$ | $IP_b$ |  | $Service_n$ |  |

Large networks which contain different platforms, operating systems, and several types of connectivity have security issues that inevitably are not noticed by the security admins.

The model we propose is a Stackelberg game model where the defender takes the first action and the attacker follows. Adding honeypots helps the defenders secure the network and creates a complex environment for the attacker. Assuming the attacker is using a scanning tool, the real host cannot be identified from the honeypots and has to search for an optimal attack plan.

For simplicity, we must consider that two hosts are identical if they run the same services and are equal in terms of connectivity.

Notations:

$N$—represents a network with services/hosts;
$s$—represents a service/host;
$T$—represents types of services/hosts;
$T = [t_1, t_2, t_3, \ldots, t_n]$, where $t$ represents a type of service/host;
$S = [s_1, s_2, s_3, \ldots, s_n]$—hosts/services of type $t$ in network $N$;
$H = [h_1, h_2, h_3, \ldots, h_n]$—honeypots of type $t$ in network $N$, where $H \subset N$;
$A = [a_1, a_2, a_3, \ldots, a_n]$ represents the attacker's actions;
$D = [d_1, d_2, d_3, \ldots, d_n]$ represents the defender's actions.

Placing several honeypots into our network can represent the defender's actions and the network is represented by:

$$d(t) = h(t) + s(t) \tag{1}$$

The probability that an attacker chooses a honeypot increases with the number of honeypots deployed in the network. If the attacker has access to a honeypot, the defender can be alerted, and the attack should be blocked with 0 second of network downtime.

Everything resumes to costs and success probabilities so we must define the following:

$c(s)$—the cost of the service/host which is duplicated with a honeypot;
$l(s)$—the loss of the service and data;
$c(a)$—the cost of the attacker;
$c(d)$—the cost of the defender.
$p(a)$—the attacker's success probability;
$resp(a)$—the attacker's response to the defender's action;
$resp(d)$—the defender's response to the attacker's action;

The honeypots are defined in our work as fake host that resembles real hosts, running services that resemble the real ones. The necessity of using honeypots is to create a more complex environment where the defender has more time and can gather more information about an attack. The work is focused on the defender as the first player in our game.

The cost of installing and maintaining a honeypot depends on the type of real host. Shandilya et al. stated in their work [13] that the computational systems became more complex, were neither fully controllable nor predictable and exposed the system behavior into nondeterministic spaces.

To have a probabilistic approach, we propose a course-of-action (COA) graph in which every path leads to an undesirable state based on security issues found in every host.

For the analysis, we use the dependency attack graph in Figure 1 in which the nodes are separated into fact nodes F (OR) and action nodes A (AND), and the following relations are established:

1.  For every action, there is a precondition and the actions of an attacker must meet vulnerabilities inside F nodes;
2.  In order for the action to be performed the facts must be true so the attacker can exploit the vulnerability and gain access to the following node.

The logical structure of our network is represented by facts nodes, and the attacker's behavior is represented by action nodes. The path with the highest reward would be chosen by the attacker.

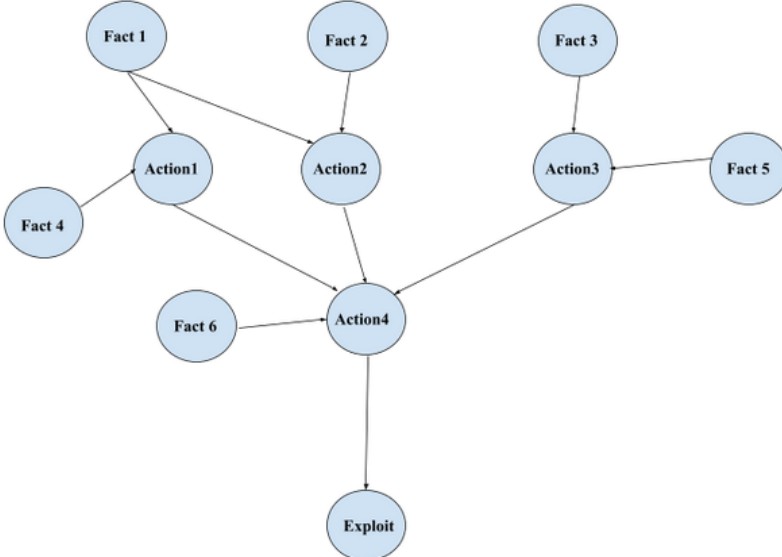

**Figure 1.** Attack graph.

The relationships between vulnerabilities that are exploitable on each node describe an attack graph and as in [6], the nodes represent a state (e.g., host, privilege and vulnerability). An attack graph (AG) illustrates the relationships among various vulnerabilities exploitable by an attacker and the privileges obtainable by the attacker. Depending on the representations of nodes and edges, different AGs can be generated.

As stated previously, two hosts who run the same services are considered identical, so the interaction with an identical host will take place because it increases the possibility to find a honeypot and the game will end.

In Figure 1, for every fact that is true, there is an action that has a probability and cost depending on the attack host's type.

*3.3. Problem Formulation*

In order to reduce their costs for the network protection, the attacker's actions *a* and defender's actions *d* have opposite performance goals because the performance cost must increase for the attacker and decrease for the defender. The payoffs *G* for attacker and defender are $G^a(a, d)$ and $G^d(a, d)$, resulting in a zero-sum game:

$$G^a(a, d) = -G^d(a, d) \tag{2}$$

Being a leader in this game, the defender can be first in making decisions and setting up the costs. The Stackelberg equilibrium (*SE*) is obtained when the defender's action *d* is taken into consideration by the attacker's action *a* to give a response.

We propose **course-of-action Stackelberg game (CoASG)** using the cost of each side and the game process.

In order to model the attacker's decisions and actions, complementary to the gathered network information, the following can be taken into consideration:

1. The attacker can terminate their attack at any moment by themselves;
2. When the attacker is interacting with a honeypot, the probability of success is

$$p_1 = 1 - \sum_{t \in T} \frac{a_t - h_t}{a_t} \tag{3}$$

3. When there is no interaction with a honeypot nor success, the probability is

$$p_3 = 0 \tag{4}$$

Using the backward induction algorithm, we can find a Stackelberg equilibrium. There can be multiple SEs, but using the costs of each party, the SE with the lowest cost can be chosen.

Phase 1. (a) The attacker takes into consideration the defender's *d* action and responds accordingly:

$$resp(d) = argmax_a G^a(a, d) \tag{5}$$

(b) The best course of action that is chosen by the attacker if there are multiple SEs is the one with the lowest cost:

$$resp_0(d) = argmin_{resp(d)} \left( \sum_{i=1}^{n} c(a)_i resp(d)_i \right) \tag{6}$$

Phase 2. (a) From the defender's perspective, the course of action is chosen after the attacker's response by maximizing its payoff:

$$resp(a) = argmax_d G^d(resp_0(d), d) \tag{7}$$

(b) Choosing the SE with the lowest cost gives the new course of action:

$$resp_0(a) = argmin_{resp(a)} \left( \sum_{i=1}^{n} c(d)_i resp(a)_i \right) \tag{8}$$

The defender's cost for adding honeypot $h(t)$ inside the network is

$$c_d(h) = \gamma l_d(h) \tag{9}$$

where $\gamma \in \mathbb{R}^+$ is a parameter that can be altered.

Some authors addressed the problem of honeypot allocation over attack graphs in different ways. The term cyberdeception game is a growing topic as stated in [14], and the art of camouflaging is applicable in network security as well.

In [15], the authors stated that securing a computing infrastructure was extremely costly and there was a clear demand for developing an automated decision support system that came to the help of a security administrator to configure the defense of the network and detect eventual attacks. They used the same technique for strengthening the network with honeypots and using them as decoys for intruders. In the paper [15], the authors proposed a model where the attacker knew the number of fake honeypots but did not know the services running on these honeypots. This assumption did not help the defender to fulfill their role but the attacker already knew what to attack and the graph was not very conclusive. Using probabilistic metrics that can be read from the Common Vulnerability Scoring System [16], some authors [15] developed an attack policy that characterized the attacker and the likelihood of a situation occurring inside the network, where the optimal state was reached when the attacker reached the maximal reward.

Our method proposes that the cost of each node in the attack graph is defined as a ratio of the node cost and how many vulnerabilities coexist on that node. The cost of the first node is distributed to the following nodes based on some rules till it reaches the targeted node to be defended. Therefore, the initial hardening is focused on the first node with the greatest impact. Even if the attacker is using some tools for reconnaissance, not all the information about the hosts and the services which are running on these hosts can be found. Even if there is no possibility to know all the vulnerabilities that coexist on a specific host, the defender can use the MITRE ATT&CK framework with the known vulnerabilities and misconfiguration status. The ATT&CK framework can provide a status report and help the defender better understand the network issues and decide whether to focus on a misconfiguration, security issue or other vulnerabilities [17]

Our approach is to adopt the partially observable Markov decision process (POMDP) so that we focus on the defender's strategy to deceive the attacker and try to maximize their reward. As stated previously, using scanning tools, the attacker can gain some knowledge about the network in the reconnaissance state, and we can assume that each player can observe the partial state. Thus, we propose a solution for the partially observable stochastic game using the partially observable Markov decision process.

Notations:

$V$ represents a vulnerability, where $V = [V_1, V_2, V_3, \ldots, V_n]$;
$F$ represents all the states in the action space, where $F = [f_1, f_2, f_3, \ldots, f_n]$;
$O^a$ represents the attacker's observations, where $O^a = [o_1^a, o_2^a, o_3^a, \ldots, o_n^a]$;
$O^d$ represents the defender's observations, where $O^d = [o_1^d, o_2^d, o_3^d, \ldots, o_n^d]$;
$B$ represents the belief states, where $B = [b_1, b_2, b_3, \ldots, b_n]$.

A vulnerability is represented by a node $V$ and the edge connecting two nodes $V_1$ and $V_2$ leads us to the statement that a second vulnerability represented by a second node $V_2$ can be exploited only if the first vulnerability $V_1$ is exploited. On their side, the attacker decides which node to attack by exploiting the vulnerability, and on the defender's side, they decide how many honeypots to use and where to deploy them along the graph edge.

At this stage, we introduce $O^a$ as an attacker's observation set and $O^d$ as a defender's observation set and

$$Pr(o_n^a, o_n^d, f_n | f_1, o_1^a, o_1^d) \tag{10}$$

as being the probability of transition from state $f_1$ to state $f_n$ while observing $O^a, O^d$ under actions $A, D$.

$G^a(f,d,a)$ represents the attacker's payoff in state $f$, so we can assume that

$$V(b_n) = max_{a_n}[G^a(a_n, f_n) + \sum_{f \in F} \tau(f_1, d, f_n)V(f_1)] \qquad (11)$$

Introducing in (11) the probable state update function $\tau(f_1, d, f_n)$ connects the belief space to the action space and represents the core component of the function.

The state transition and the observation of each transition must be known to calculate this function, and the defender knows the state transition model.

Observations are connected to the attacker's actions, which denote the probability of seeing observation $O$ while transitioning from belief state $b$ to a state of belief $b(f)$.

$$O^d(o_1^d, f_1, a_1) = Pr(o_1^a | f_1, a_1, b_1) \qquad (12)$$

The defender can estimate that at a given state $f \in F$, the attacker played a specific action and calculate the possibility function as follows:

$$Pr(o_1^d | f_1, a_1, b_1) = \sum_{f \in F} Pr(o_1^d | f_1, a_1, f_1)b(f_1) \qquad (13)$$

From the survey in [7], we can see the difference between the two methods presented above by mentioning that the Markov decision process (MDP) can be applied to the visible state when we know everything about the network, but our proposed model using POMDP can be applied to the invisible state when an opponent in a game must observe, collect information and then act. Because the defender cannot be sure about a particular state, they need to determine in which state they are in by perceiving the environment and the concept of the belief state space can be introduced.

## 4. Results

### 4.1. Modeling the System

Let us consider some notations to help us understand the model. On every host, there were multiple running services which can be real ($\lambda_{real}$) or honeypot ($\lambda_{honeypot}$) services. In our model, we took several hosts, which could be considered as servers, on top of which were configured running services such as web services, database services, file host services, etc. Among the real services, there were honeypots that served as decoys for attackers.

The model is presented in Figure 2 where the access to the cloud (outside) is made through a router, firewalls protect the inside network with applied rules, and on every physical machine, there are running multiple virtual machines (hosts). These hosts can be real ones or honeypots, but the mirroring is made for the ones with high interest.

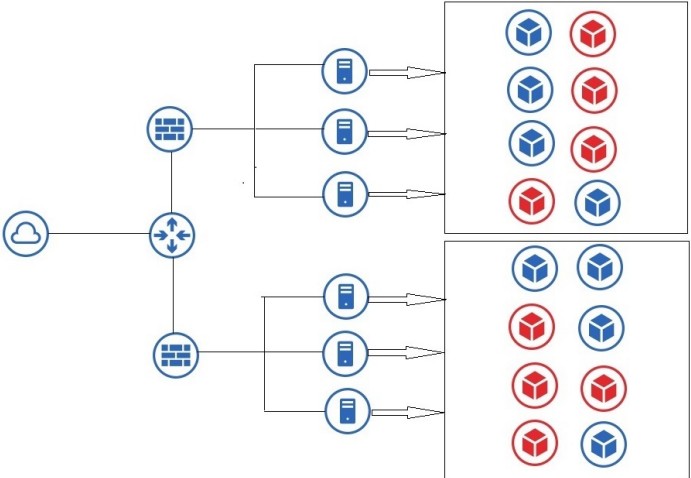

**Figure 2.** System model representation.

In our system, we used notations to describe the services which could be in three states at any time:

**State 1.** The service $(\lambda)$ is running/opened $(\lambda_{real})$.
**State 2.** The service is a honeypot $(\lambda_{honeypot})$.
**State 3.** The service is a closed $(\lambda_{closed})$.

The multitude of services in every state that is on a host or server can be noted with

$$\Lambda = \{\lambda_1, \lambda_2, \ldots, \lambda_n\} \tag{14}$$

but because of the honeypots, the services become

$$\Lambda = \{\lambda_{11}, \lambda_{12}, \ldots, \lambda_{1n}\} \tag{15}$$

In our system, there were several services provided which could be in the four states:

**State 1.** $\lambda_{10}$ the service is closed;
**State 2.** $\lambda_{11}$ the service is opened;
**State 3.** $\lambda_{20}$ the user from the network cannot access the service;
**State 4.** $\lambda_{21}$ the user from the network can access the service.

To define a simple strategy, let us take into consideration that a service can be real, a honeypot or closed, then from (15), we can decide when the attacker can access the services

$$\lambda = \{\lambda_{111}, \lambda_{121}, \ldots, \lambda_{1n1}\} \tag{16}$$

and when the attacker cannot access the services

$$\lambda = \{\lambda_{110}, \lambda_{120}, \ldots, \lambda_{1n0}\} \tag{17}$$

From (16) and (17), it results that the simple strategy $\alpha$ is

$$\alpha = \{\lambda_{111}, \lambda_{110}\} \tag{18}$$

To calculate the attacker's and defender's payoffs, we must define the following parameters with conditions

$G^d > 0$;
$C^a$ is the cost of the attacker and has the relation $G^d \geq C^a > 0$;
$G^h > 0$ is the honeypot's payoff;
$\gamma \geq 1$ the damage factor of the attacker;
$\eta \geq 1$ is the deceive factor of using a honeypot.

We can calculate the payoffs for the two cases:
**Case 1.** The defender is providing a real service on a host and the attacker can access it; then, the payoffs are

$$G^a = \gamma G^d - C^a \tag{19}$$

and

$$G^d = -\gamma G^a \tag{20}$$

**Case 2.** The defender is providing a fake service and the payoffs are

$$G^a = -\eta G^h - C^a \tag{21}$$

and

$$G^d = \eta G^h \tag{22}$$

Having a system with $s$ services/hosts can result in the payoff matrix in Table 2.

**Table 2.** Payoff matrix.

| | | Attacker | | Defender | |
|---|---|---|---|---|---|
| | | $\lambda_{21}$ | $\lambda_{20}$ | $\lambda_{21}$ | $\lambda_{20}$ |
| $\lambda_{110}$ | $\lambda_{11}$ | $(-\gamma G^d/s, \gamma G^d/s - C^a)$ | (0,0) | $(G^d, G^d)$ | (0,0) |
| | $\lambda_{10}$ | $(0, -C^a)$ | (0,0) | $(-G^d, -G^d)$ | (0,0) |
| $\lambda_{111}$ | $\lambda_{11}$ | $(\eta G^h/s, -\eta G^d/s - C^a)$ | (0,0) | $(0, -G^d)$ | (0,0) |
| | $\lambda_{10}$ | $(0, -C^a)$ | (0,0) | $(0, -G^d)$ | (0,0) |

Using the payoff matrix described in Table 2, we can state that we have a $m * m$ matrix $X = (x_{ij})_{mm}$ where $x_{ij}$ is represented by the row player's payoff when the $i$th strategy is picked while the column player chooses the $j$th strategy, and in general, we can write the following payoff matrix

$$\begin{bmatrix} -\gamma G^d/s & \gamma G^d/s - C^a & 0 & 0 \\ 0 & -C^a & 0 & 0 \\ \eta G^h/s & -\eta G^d/s - C^a & 0 & 0 \\ 0 & 0, -C^a & 0 & 0 \end{bmatrix} \tag{23}$$

### 4.2. Evaluating the System

For the simulation, we focused on the game between the attacker and the defender; we used Gambit V15.1.1, which is a library of game theory software and tools for the analysis of finite extensive and strategic games [18], and MATLAB R2021b v9.11.0.

The parameters used for running the simulation are defined in Table 3.

**Table 3.** Simulation parameters.

| Parameter | Value | Observation |
|---|---|---|
| $G^d$ | 50 | Defender's payoff |
| $C^a$ | 40 | Attacker's cost |
| $G^h$ | 40 | Honeypot's payoff |
| $\gamma$ | 2 | Damage factor |
| $\eta$ | 1 | Deceive factor |
| $N_1$ | 1 | A network with 1 service/host |
| $N_{10}$ | 10 | A network with 10 services/hosts |
| $N_{50}$ | 50 | A network with 50 services/hosts |
| $N_{100}$ | 100 | A network with 100 services/hosts |

Using Gambit to simulate one service/host and applying the payoff matrix in (23), we could observe that the attacker had a real advantage and predominance, suggesting that the defender could suffer great losses (Figure 3).

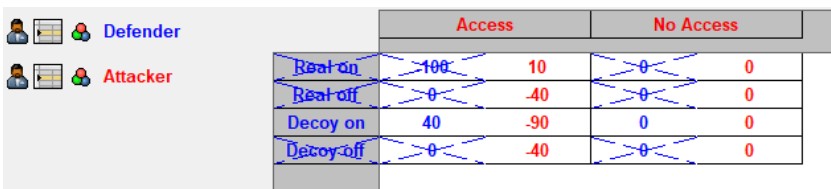

**Figure 3.** Results of the dominance for $N_1$.

We can notice from Figures 4–6 that when the number of servers or hosts increased and the resources available for honeypots were usable, the results showed that the dominance was transferring to the defender's side and the attacker could not easily access the network anymore.

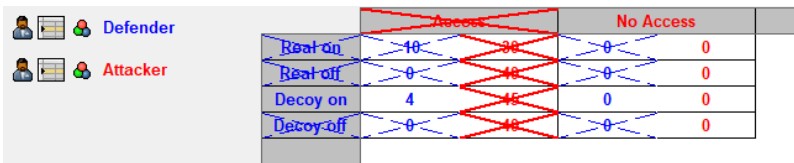

**Figure 4.** Results of the dominance for $N_{10}$.

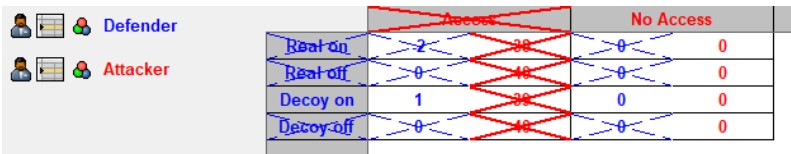

**Figure 5.** Results of the dominance for $N_{50}$.

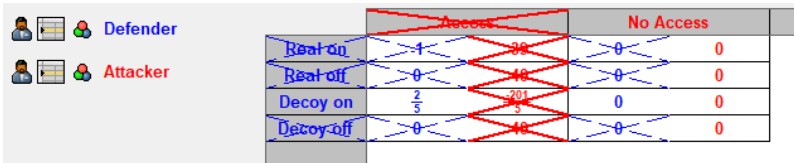

**Figure 6.** Results of the dominance for $N_{100}$.

We can notice that when the number of services/hosts increased, the payoff of the defender decreased, being influenced by the honeypots' costs. Furthermore, the attacker's payoff decreased at the same time as the increase in the defender's payoff, and the cost of attacking the network rapidly grew to the point where it was very risky to be caught, or too much time was consumed.

In a big network, there can be more than 1000 hosts or services, but applying our method of placing honeypots, we can conclude that 50 honeypots are enough for a defense strategy. There is no need to deploy more honeypots and all the data that are gathered can be used in order to mitigate an attacker's infiltration in the system. The purpose of the experiment was to show the use of honeypots to increase the security of a system, but the question was to know when a system administrator can be assured that the number of fake nodes can help them out in protecting the network with fewer costs.

Using the extensive (tree) game from Gambit v15.1.1, we introduced the values obtained in Figures 3–6 and then calculated the Nash equilibrium and the results shown in Figures 7–10 demonstrate that our proposed method had a great output for the defender. The honeypots decreased the attacker's dominance and increased the chances that the defender would observe and stop their actions.

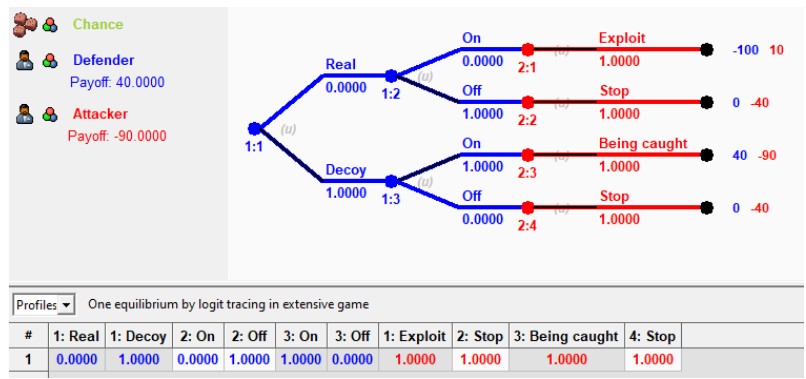

**Figure 7.** Results of the dominance for $N_1$.

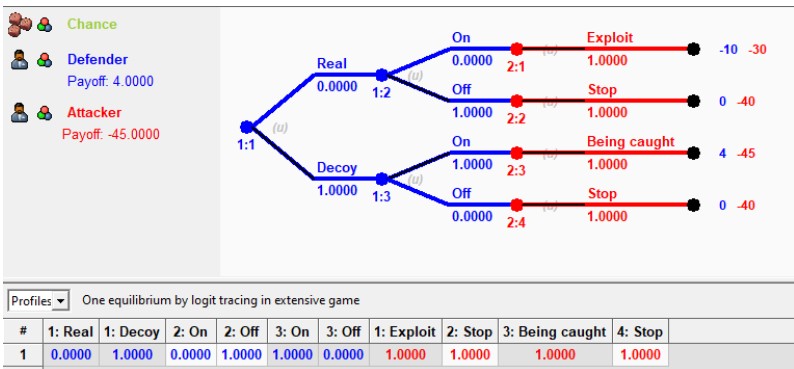

**Figure 8.** Results of the dominance for $N_{10}$.

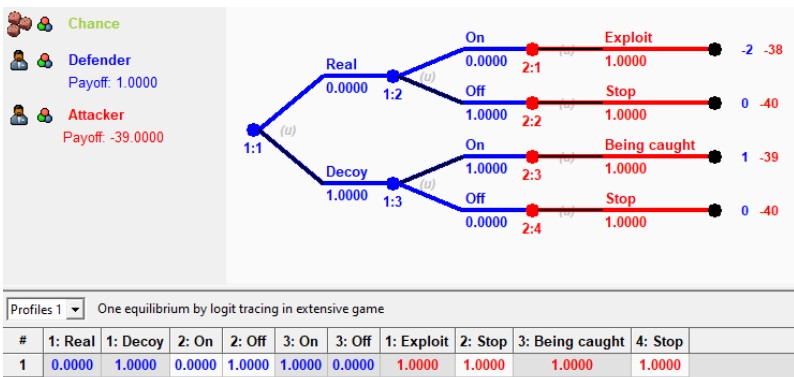

**Figure 9.** Results of the dominance for $N_{50}$.

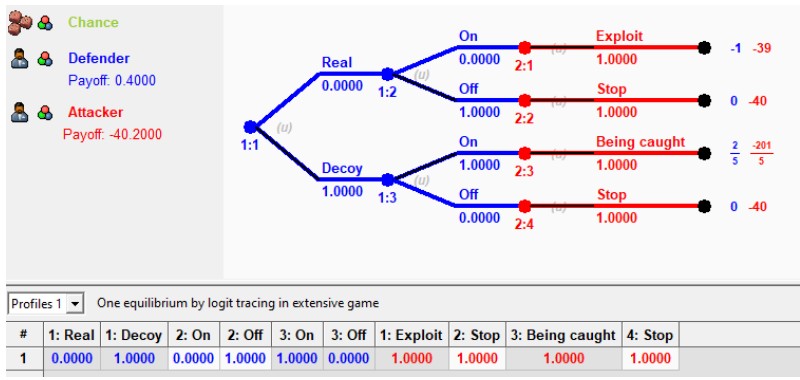

**Figure 10.** Results of the dominance for $N_{100}$.

The Nash equilibrium is the strategy with a high probability that an attacker would take into consideration, while the defender is trying to solve the imperfect game with less information. The defender places the honeypots based inside the network and the attacker tries to avoid them and not being spotted. Both players are paying some costs, the defender for placing the honeypots and the attacker for trying to evade them and access real hosts. We found out that the game could come to an idle phase when both attacker and defender responses $resp(d)$ and $resp(a)$ were equal to 0. This meant that both persons in that game were not making any moves and were in idle mode.

In our case, the reward of the defender increased when the attacker was caught, and the capture cost also increased. However, for a large network with more than 50 services/hosts, the cost of deploying honeypots is very high, but it forces the attacker to step back to avoid high-risk actions; therefore, the defender's reward decreases, but the network is safe.

The comparison between our algorithm and a fixed policy for honeypot allocation shows that this last one does not recognize the dynamics of the game and is not flexible because all the information is gathered at the beginning of the game.

Now, using the values from Table 3, we can calculate the optimal strategy using (23)

$$
\begin{bmatrix}
-100 & 10 & 0 & 0 \\
0 & -40 & 0 & 0 \\
40 & -90 & 0 & 0 \\
0 & -40 & 0 & 0
\end{bmatrix}
\tag{24}
$$

To solve this payoff matrix and use the algorithm developed in [19], we must convert the matrix to have no negative entries by adding a suitable fixed number to all the entries in the matrix. Let us add 110, and the new matrix is

$$
\begin{bmatrix}
10 & 120 & 110 & 110 \\
110 & 70 & 110 & 110 \\
150 & 20 & 110 & 110 \\
110 & 70 & 110 & 110
\end{bmatrix}
\tag{25}
$$

Thus, solving two linear programming problems determines the optimal strategies

$$
min(x_1 + x_2 + x_3 + x_4)
\tag{26}
$$

$$
\begin{cases}
10x_1 + 110x_2 + 150x_3 + 110x_4 \geq 1 \\
120x_1 + 70x_2 + 20x_3 + 70x_4 \geq 1 \\
110x_1 + 110x_2 + 110x_3 + 110x_4 \geq 1 \\
110x_1 + 110x_2 + 110x_3 + 110x_4 \geq 1
\end{cases}
$$

and

$$
max(y_1 + y_2 + y_3 + y_4)
\tag{27}
$$

$$
\begin{cases}
10y_1 + 120y_2 + 110y_3 + 110y_4 \leq 1 \\
110y_1 + 70y_2 + 110y_3 + 110y_4 \leq 1 \\
150y_1 + 20y_2 + 110y_3 + 110y_4 \leq 1 \\
110y_1 + 70y_2 + 110y_3 + 110y_4 \leq 1
\end{cases}
$$

To calculate in Matlab the optimal strategy, we translated the linear program function into the format

$$
min(x_1 + x_2 + x_3 + x_4)
$$

$$
\begin{cases}
-10x_1 - 110x_2 - 150x_3 - 110x_4 \leq -1 \\
-120x_1 - 70x_2 - 20x_3 - 70x_4 \leq -1 \\
-110x_1 - 110x_2 - 110x_3 - 110x_4 \leq -1 \\
-110x_1 - 110x_2 - 110x_3 - 110x_4 \leq -1
\end{cases}
$$

To find the solution, we used the *linprog* function from the optimization toolbox in Matlab so

$$
c = [1, 1, 1, 1];
$$
$$
a = [-10, -110, -150, -110; -120, -70, -20, -70; -110, -110, -110, -110; -110, -110, -110, -110];
$$
$$
b = [-1, -1, -1, -1];
$$
$$
lb = [0, 0, 0, 0]
$$

Applying the formula $x = linprog(c, a, b, [], [], lb)$, we found the optimal solution as

$$
x_r = (0.0032, 0, 0, 0.0088)
\tag{28}
$$

and for the second LP, we used the translation of (25) and obtained

$$min(-y_1 - y_2 - y_3 - y_4)$$

$$\begin{cases} 10y_1 + 120y_2 + 110y_3 + 110y_4 \leq 1 \\ 110y_1 + 70y_2 + 110y_3 + 110y_4 \leq 1 \\ 150y_1 + 20y_2 + 110y_3 + 110y_4 \leq 1 \\ 110y_1 + 70y_2 + 110y_3 + 110y_4 \leq 1 \end{cases}$$

c = [−1, −1, −1, −1];
a = [10, 120, 110, 110; 110, 70, 110, 110; 150, 20, 110, 110; 110, 70, 110, 110];
b = [1, 1, 1, 1];
lb = [0, 0, 0, 0]

so after applying in Matlab the command $y = linprog(c, a, b, [], [], lb)$, we obtained

$$y_r = (0.004, 0.008, 0, 0) \tag{29}$$

After applying the *linprog* function in Matlab for 1, 10, 50, and 100 service(s)/host(s) in our environment using the method described earlier, we calculated the payoff matrix. The Matlab simulation results presented in Figure 11 show that the costs for both players increased significantly with N increasing.

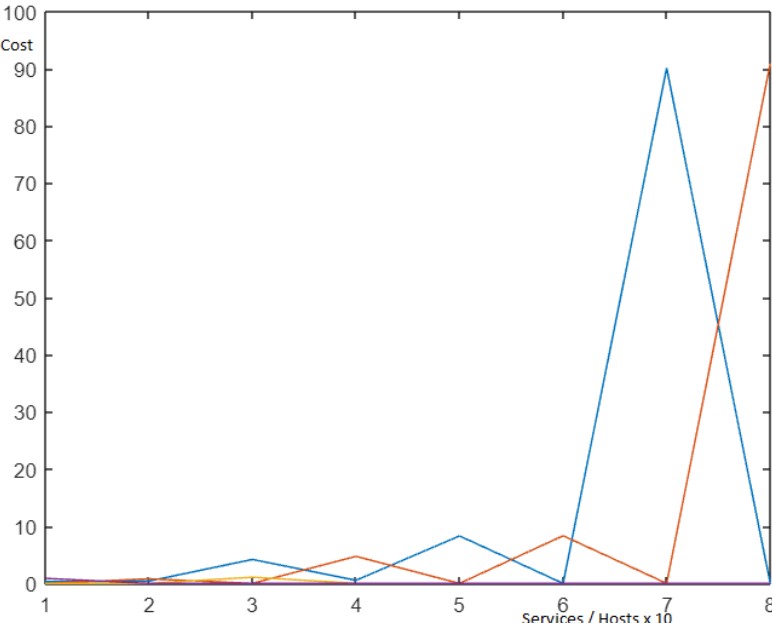

**Figure 11.** Linear programming matrix results.

With more than 50 honeypots, the costs for both players increase and the probability of the attacker being caught is extremely high, so using a defense strategy with only 50 honeypots can be applied successfully.

## 5. Conclusions

In this paper, we proposed a solution for keeping real services/hosts safe by deploying honeypots. The game-theoretic approach and analysis verified in Matlab and Gambit demonstrated the effectiveness of our model but also compared our results with some existing models such as fixed allocation. The equilibrium in the zero-sum game proposed adjusted the allocation of honeypots based on probabilities. With more than 50 honeypots,

the costs for both players increased, the attacker lost much time and the probability of being caught was extremely high. Even if the network was bigger and had more than 1000 real hosts, a defense strategy could be applied using only 50 honeypots. Another interesting result was that the defender's and attacker's payoffs were influenced by the number of nodes and in a network with more than 50 honeypots, the costs for both players were increasing rapidly and were not sustainable anymore. This method of protection can help even if the attacker is using some techniques such as machine learning algorithms in order to detect honeypots. Huang [8] developed a detection algorithm for honeypots but the costs for the attacker would still increase and this aspect should be addressed in future work in order to determine the strength of our model when faced with a machine learning algorithm.

Further research is needed to test our model in a real lab environment and map some attack techniques to improve the model.

**Author Contributions:** All authors have contributed equally to this manuscript. All authors have read and agreed to the published version of the manuscript.

**Funding:** This research received no external funding.

**Institutional Review Board Statement:** Not applicable.

**Informed Consent Statement:** Not applicable.

**Data Availability Statement:** The data used to support the findings of this study are available from the corresponding author upon request.

**Conflicts of Interest:** The authors declare no conflict of interest.

## Abbreviations

The following abbreviations are used in this manuscript:

| | |
|---|---|
| CCP | Cloud computing providers |
| ENISA | European Union Agency for Cybersecurity |
| DOS | Denial of service |
| CoA | Course of action |
| CoASG | Course-of-action Stackelberg game |
| CVSS | Common Vulnerability Scoring System |
| POMDP | Partially observable Markov decision process |
| MDP | Markov decision process |

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
