# Peer review of "A Game-Theoretic Approach for Network Security Using Honeypots"

_futureinternet, doi:10.3390/fi14120362_

Round 1
Reviewer 1 Report
In this paper, the authors propose a method to solve a game theory model by defining the contents of the game payoff matrix and incorporating honeypots in defence strategy. The paper elaborates sufficiently the approach adopted in incorporating the defense strategy of increasing the honeypots, and evaluating the cost to the attacker. The experimental results sufficiently demonstrate the outcome of the proposed approach.
Author Response
Cover Letter
A Game Theoretic Approach for Network Security using Honeypots
Răzvan Florea and Mitică Craus
Dear Reviewer,
Thank you for your time and patience in analyzing our paper and we value your trust. Finally, we appreciate the time and work you devoted to improving the quality of our study.
Sincerely,
Florea Razvan
On behalf of all authors
November 18, 2022
“Gheorghe Asachi” Technical University, Iasi, Romania
Faculty of Automatic Control and Computer Engineering
Department of Computer Science and Engineering

Reviewer 2 Report
The authors propose Course of Action Stackelberg Game (CoASG) to solve a game theory model that includes the use of honeypots.
The overall paper is well written and easy to read. The paper seams lacking in terms of comparison with related work. Aside from two paragraphs in the end of section 2.1, I was unable to find such comparison. For instance, the papers [1],[2],[3],[4] and [5], all recently published, are related and could be compared.
The presented work does not appear to be a significant improvement over state of the art. Sections 2 and 3 are well presented. I failed to understand how the honeypot detection and avoidance capability of the attacker was included in this work. If an attacker can detect a honeypot with a relevant degree of certainty[6], it can also ignore it, favoring other attack paths.
Finally, the deployment of 50 honeypot services requires relevant resources, making it impossible to use in a multitude of scenarios.
[1] Khouzani, M. H. R., Liu, Z., & Malacaria, P. (2019). Scalable min-max multi-objective cyber-security optimisation over probabilistic attack graphs. European Journal of Operational Research, 278(3), 894-903.
[2] Abdallah, M., Naghizadeh, P., Hota, A. R., Cason, T., Bagchi, S., & Sundaram, S. (2020). Behavioral and game-theoretic security investments in interdependent systems modeled by attack graphs. IEEE Transactions on Control of Network Systems, 7(4), 1585-1596.
[3] Yoon, S., Cho, J. H., Kim, D. S., Moore, T. J., Free-Nelson, F., & Lim, H. (2020). Attack graph-based moving target defense in software-defined networks. IEEE Transactions on Network and Service Management, 17(3), 1653-1668.
[4] Lallie, H. S., Debattista, K., & Bal, J. (2020). A review of attack graph and attack tree visual syntax in cyber security. Computer Science Review, 35, 100219.
[5] Zeng, J., Wu, S., Chen, Y., Zeng, R., & Wu, C. (2019). Survey of attack graph analysis methods from the perspective of data and knowledge processing. Security and Communication Networks, 2019.
[6]Huang, C., Han, J., Zhang, X., & Liu, J. (2019). Automatic identification of honeypot server using machine learning techniques. Security and Communication Networks, 2019.
Author Response
Cover Letter
A Game Theoretic Approach for Network Security using Honeypots
Răzvan Florea and Mitică Craus
Dear Reviewer,
We appreciate your suggestions that have contributed to improving our document. In the new version of the paper, we tried to improve its content and form based on your suggestions. All the changes in the body of the paper have been highlighted in blue color to be easily identified by the reviewers and editor.
Comment 1: The paper seems lacking in terms of comparison with related work.
Response 1: In the reviewed version of the paper, there was included a Related Work section in order to provide a comparison with other related works. The papers mentioned by you were studied and were taken into consideration for comparison. All authors had different approaches from shuffling techniques of IPs in Software Defined Networks for Cloud Infrastructures to using Machine Learning Algorithms in order to identify honeypots. All the authors concluded the necessity of standardization of Attack Graphs in network topologies and the real benefits of this method in providing the security of a network.
Comment 2: I failed to understand how the honeypot detection and avoidance capability of the attacker were included in this work. If an attacker can detect a honeypot with a relevant degree of certainty, it can also ignore it, favoring other attack paths.
Response 2: The attacker can use some techniques to identify the honeypots and some authors introduce an Artificial Intelligence method to identify honeypots, but our model is focused on the defender. As stated in lines 182-184 the defender is a leader in the game and can be first in taking decisions. As you stated correctly in your comment, the honeypot detection, and avoidance capability were not included in this work, but it is meant to be studied furthermore in a future paper to test out the framework from the attacker’s point of view.
Finally, we appreciate the time and work you devoted to improving the quality of our study and hope that all the corrections and suggestions that were applied to the final version of the paper meet your requirements.
Sincerely,
Florea Razvan
On behalf of all authors
November 18, 2022
“Gheorghe Asachi” Technical University, Iasi, Romania
Faculty of Automatic Control and Computer Engineering
Department of Computer Science and Engineering

Reviewer 3 Report
The paper presents a game theory model of the interaction between an attacker and a defender for a system using honeypots in terms of the game payoff matrix. They use a Course of Action (CoA) Stackelberg Game (CoASG) based approach.
The objective of the paper is not clear. Although the paper states that the objective is to "find the best possible combination of the strategies", it is not clear if it from the attacker's perspective or the defender's perspective. The paper claims to use a probabilistic approach, but I could not see how the probabilities are used.
The paper is hard to follow since many of the key terms and notation used are not defined properly.
Table 1: VMa, VMb etc: are these virtual machines?
"N - represents a network with services/hosts;" should perhaps be "n - number of services/hots in the network"
Are the honeypots services or hosts?
The equation 1 is not clear.
d(t) = h(t) + s(t)
Are h(t), s(t) binary or integers? What does the "+" operation mean?
"facts nodes F (OR) and action nodes A (AND)" what are facts and actions?
How are equations 3 and 4 obtained? Are they taken from a reference?
"cost of each node in the attack graph is defined as a ratio of node cost and how many vulnerabilities coexist on that node" this is not clear. At any given point in time, a number of vulnerabilities exist that are not yet known to exist.
Figures and tables are not discussed. For example Fig 2 is not described.
Table 3: what is "damage factor".
Fig 3-10 are not described.
Fig 11: Services/hosts vs cost: Why is there a peak at 70 and 80, and why cost drop down 0 after that?
The paper needs to be carefully rewritten for a reader to be able to follow what the authors are trying to achieve.
Author Response
Cover Letter
A Game Theoretic Approach for Network Security using Honeypots
Răzvan Florea and Mitică Craus
Dear Reviewer,
We appreciate your suggestions that have contributed to improving our document. In the new version of the paper, we tried to improve its content and form based on your suggestions. All the changes in the body of the paper have been highlighted in blue color to be easily identified by the reviewers and editor.
Comment 1: The objective of the paper is not clear. Although the paper states that the objective is to "find the best possible combination of the strategies", it is not clear if it is from the attacker's perspective or the defender's perspective.
Response 1: In our paper, the contribution was to transform the focus on the defender’s problem and try to provide a linear programming problem. The honeypots which are added to the defender’s strategy are considered mirroring real hosts on top of which are running services. As we stated earlier the defender is a leader in the game and can be first in taking decisions.
Comment 2: The paper is hard to follow since many of the key terms and notations used are not defined properly.
Response 2: The key terms and notation were analyzed and defined properly like VMa is a Virtual Machine on top of which are running services so the honeypots are considered hosts on top of which are running services.
Comment 3: Are the honeypots services or hosts?
Response 3: As we stated before the honeypots in the defender’s strategy are considered to mirror real hosts on top of which are running services.
Comment 4: “facts nodes F (OR) and action nodes A (AND)” what are facts and actions?
Response 4: The logical structure of our network is represented by facts nodes (OR) which represent vulnerabilities, and the attacker behavior is represented by action nodes (AND). The relationships between vulnerabilities that are exploitable on each node describe an attack graph and the nodes represent a state (e.g. host, privilege, and vulnerability).
Comment 5: Figures and tables are not discussed. For example, Fig 2 is not described.
Response 5: As you stated and were right, we reviewed the paper and the tables and figures were described properly and discussed.
Comment 6: Fig 11: Services/hosts vs cost: Why is there a peak at 70 and 80, and why did cost drop down to 0 after that?
In Figure 11, the results obtained after the Matlab simulation show the strategy of using more than 50 honeypots increases the costs for both defender and attacker. The results obtained in Matlab show that the cost is bouncing for both players and increases rapidly after 50 honeypots are deployed. As stated in lines 178-181, the attacker and defender have opposite performance goals because the performance cost must increase for the defender and decrease for the attacker.
Finally, we appreciate the time and work you devoted to improving the quality of our study and hope that all the corrections and suggestions that were applied to the final version of the paper meet your requirements.
Sincerely,
Florea Razvan
On behalf of all authors
November 18, 2022
“Gheorghe Asachi” Technical University, Iasi, Romania
Faculty of Automatic Control and Computer Engineering
Department of Computer Science and Engineering

Reviewer 4 Report
The use of the model to increase safety is not in doubt, but it probably does not fit well with the stages of the attack. The best example of a cyber-stage attack is MITRE. The provided source does not reflect the stage of the attack. However, this is not a major drawback, just an observation. It is great that all of Matlab's features were used, and that the simulation was done. This kind of model verification can now be published.
The work only partially reflects the goal of ensuring preventive cyber and network security, but this does not in the least diminish the effect of the experiment and science. Such work may be published publicly.

Author Response
Cover Letter
A Game Theoretic Approach for Network Security using Honeypots
Răzvan Florea and Mitică Craus
Dear Reviewer,
Thank you for your time and patience in analyzing our paper and we value your trust.
Comment 1: The provided source does not reflect the stage of the attack.
Response 1: You are absolutely right that our model does not reflect the stage of the attack but we think it can be used with MITRE ATT&CK Framework in securing a network environment.
Finally, we appreciate the time and work you devoted to improving the quality of our study.
Sincerely,
Florea Razvan
On behalf of all authors
November 18, 2022
“Gheorghe Asachi” Technical University, Iasi, Romania
Faculty of Automatic Control and Computer Engineering
Department of Computer Science and Engineering
